# Structure and assembly of scalable porous protein cages

Eita Sasaki[1], Daniel Böhringer[2], Michiel van de Waterbeemd[3], Marc Leibundgut[2], Reinhard Zschoche[1], Albert J.R. Heck[3], Nenad Ban[2] & Donald Hilvert[1]

Proteins that self-assemble into regular shell-like polyhedra are useful, both in nature and in the laboratory, as molecular containers. Here we describe cryo-electron microscopy (EM) structures of two versatile encapsulation systems that exploit engineered electrostatic interactions for cargo loading. We show that increasing the number of negative charges on the lumenal surface of lumazine synthase, a protein that naturally assembles into a $\sim$1-MDa dodecahedron composed of 12 pentamers, induces stepwise expansion of the native protein shell, giving rise to thermostable $\sim$3-MDa and $\sim$6-MDa assemblies containing 180 and 360 subunits, respectively. Remarkably, these expanded particles assume unprecedented tetrahedrally and icosahedrally symmetric structures constructed entirely from pentameric units. Large keyhole-shaped pores in the shell, not present in the wild-type capsid, enable diffusion-limited encapsulation of complementarily charged guests. The structures of these supercharged assemblies demonstrate how programmed electrostatic effects can be effectively harnessed to tailor the architecture and properties of protein cages.

[1] Department of Chemistry and Applied Biosciences, Laboratory of Organic Chemistry, ETH Zürich, Zürich 8093, Switzerland. [2] Department of Biology, Institute of Molecular Biology and Biophysics, ETH Zürich, Zürich 8093, Switzerland. [3] Biomolecular Mass Spectrometry and Proteomics, Bijvoet Center for Biomolecular Research and Utrecht Institute for Pharmaceutical Sciences, Utrecht University, CH Utrecht 3584, The Netherlands. Correspondence and requests for materials should be addressed to D.H. (email: hilvert@org.chem.ethz.ch).

Hollow proteinaceous particles are widespread in nature. By providing a protective shell for molecular cargo, these nanostructures serve as multipurpose containers for the transport of viral payloads[1], biomineralization[2], protein folding[3] and degradation[4], enzyme encapsulation[5,6] and even catalysis of short metabolic sequences[7,8]. Inspired by this precedent, protein cages are being engineered in the laboratory as customized nanoreactors[9,10], delivery or display vehicles[11] and artificial organelles[12,13] for diverse applications in medicine, materials science and synthetic biology.

One highly versatile encapsulation system has been engineered from the capsid-forming enzyme lumazine synthase from the hyperthermophilic bacterium *Aquifex aeolicus* (Fig. 1a). Although the wild-type protein (AaLS-wt) adopts a flavodoxin-like $\alpha\beta$-fold[14] (Fig. 1b) that bears no structural similarity to viral capsid proteins[15–18], it spontaneously assembles into virus-like icosahedrally symmetric particles containing 60 identical subunits with a triangulation number[19] $T = 1$. In nature, these structures encapsulate riboflavin synthase, an enzyme that converts lumazine to riboflavin[20]. To bind unrelated guests, we mutated four residues that project into the lumen of the capsid to negatively charged glutamates. The resulting variant, called AaLS-neg (Fig. 1c,f), encapsulates proteins with complementary positive

charges[21]. Loading capacity was subsequently improved roughly 10-fold by directed evolution, exploiting the ability of the cage to sequester a toxic protease and thereby protect the proteome of *Escherichia coli*[22]. The best variant, AaLS-13, possesses seven additional mutations that further increase the container's net negative charge (Fig. 1d,g) and enhance its ability to bind positively charged proteins both *in vivo* and *in vitro*[22–24]. The recent discovery that the encapsulation process is rapid enough to be limited by the rate of cargo diffusion highlights the unique properties of these supercharged protein assemblies[25].

Befitting their greater packaging capabilities, the negatively supercharged AaLS variants have substantially larger cage diameters than the native structure (AaLS-13, $\sim 40$ nm; AaLS-neg, $\sim 30$ nm; AaLS-wt, $\sim 16$ nm)[14,21,22]. In analogy to viral proteins that can form icosahedral shells of varying size in response to external stimuli or cargo loading[26,27], the structural transformations observed in these[21,22] and other LS variants[28] could conceivably reflect expansion of the starting $T = 1$ particles to $T = 3$ (180-mer), $T = 4$ (240-mer) or even $T = 7$ (420-mer) icosahedral structures[29]. However, the molecular mechanisms underlying these dramatic transitions and, more specifically, how protomer structure controls protein cage size and architecture are poorly understood.

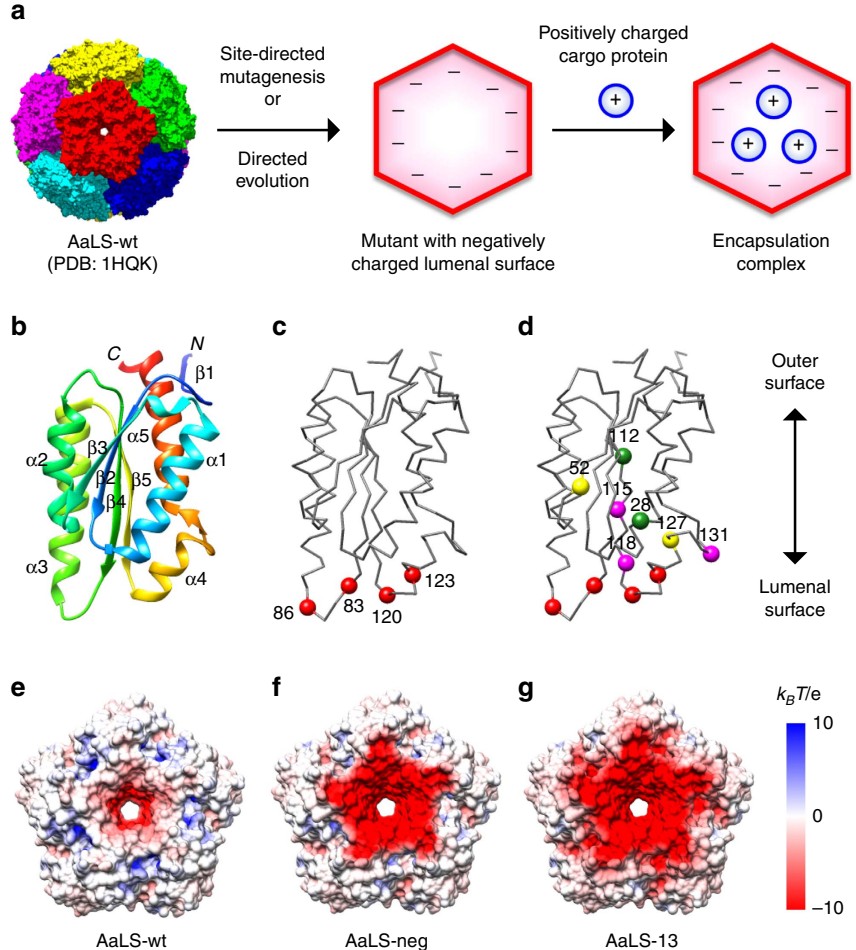

**Figure 1 | Design of the AaLS encapsulation system.** (**a**) Engineering AaLS cages to encapsulate cargo proteins based on electrostatic interactions. (**b**) A ribbon representation of the AaLS monomer. (**c,d**) Backbone traces of the AaLS-neg (**c**) and AaLS-13 (**d**) monomers in the same orientation as in **b**. Residues mutated in AaLS-neg are shown as red spheres (R83E, T86E, T120E and Q123E) (**c**), whereas the additional substitutes in AaLS-13 are shown as magenta (V115D, A118D and K131E), yellow (R52C, R127C) or green (D28G, T112S) spheres (**d**). (**e–g**) The lumenal surface of the AaLS-wt (**e**), AaLS-neg (**f**) and AaLS-13 (**g**) pentamers, showing electrostatic surface potentials assuming pH 8 and an ionic strength of 150 mM. For the variants, mutations were mapped onto the crystal structure of the wt pentamer.

Here, we elucidate the structures of the AaLS protein containers by native mass spectrometry (MS) and cryo-EM single-particle reconstruction techniques. Our results show that the expanded cages do not assemble according to well-established principles of quasi-equivalence[19] but instead adopt unprecedented topologies that preserve some pentamer interfaces and leave other edges unsatisfied. Stable gaps in these shell structures rationalize why complementarily charged guests are encapsulated so efficiently.

## Results

**Native MS analysis.** Native MS, which has provided accurate masses of viral capsids[30], was employed to determine the subunit copy numbers of the AaLS capsids (Fig. 2). As expected, AaLS-wt gave a well-resolved charge state envelope corresponding to a 60-mer (measured mass of 1.06 MDa, theoretical mass of 1.05 MDa), consistent with the previously reported $T = 1$ crystal structure[14] (Fig. 2a; Supplementary Fig. 1a,b). For AaLS-neg, a similarly well-resolved spectrum was obtained that corresponds to a 180-mer (measured mass of 3.02 MDa, theoretical mass of 3.01 MDa) in agreement with the predicted icosohedral $T = 3$ structure[21] (Fig. 2b; Supplementary Fig. 1c,d). When the latter sample was pre-incubated in a low ionic strength buffer, however, only disassembled fragments corresponding to pentamers and smaller amounts of 10-mers and 15-mers were observed (Supplementary Fig. 1e,f); no dimeric, trimeric or hexameric building blocks, which would be expected for $T = 3$ capsids[31], were detected. Although no charge states could be resolved for AaLS-13, possibly because of structural heterogeneity, the MS data suggest that this cage contains between 240 and 420 subunits (Fig. 2c).

**Single-particle reconstruction.** Encouraged by the apparent monodispersity of the AaLS-wt and AaLS-neg samples, cryo-EM single-particle reconstruction was performed to obtain detailed structural information[32]. The cryo-EM images of AaLS-wt were classified based on orientations of the particles in thin ice layers, and two-dimensional (2D) averages of each class were generated (Supplementary Fig. 2). The selected 2D class-averages were used to make an initial three-dimensional (3D) model by imposing icosahedral symmetry without any structural reference (Supplementary Fig. 2). This model was filtered to low-resolution and used as a reference for further 3D classification and refinement. The final EM density map of the AaLS-wt cage was calculated to 3.9 Å resolution from 3,268 particles (Fig. 3a), which allowed building and coordinate refinement of an atomic model (Fig. 3g; Table 1). The AaLS-wt cage is a 16-nm wide hollow dodecahedron consisting of 12 pentamers (Fig. 3d,j; Supplementary Fig. 5a). Our model agrees well with the $T = 1$ crystal structure[14] with a Cα root mean square deviation (RMSD) of 0.52 Å for all residues.

In the case of AaLS-neg, contrary to expectations for a conventional $T = 3$ state[21], imposing icosahedral symmetry during reconstruction did not result in interpretable density. By assuming tetrahedral symmetry, however, we obtained an initial reconstruction that was consistent with the distinct patterns of the 2D class-averages (Supplementary Fig. 3). This initial model was filtered to low-resolution and used for 3D reconstruction of the cage (Fig. 3b). The secondary structure elements of each subunit could be unambiguously assigned in the final EM density map refined to 5.4 Å resolution from 26,769 particles (Fig. 3h; Table 1). The 180 subunits of AaLS-neg, organized as 36 pentameric building blocks, form a novel 29-nm hollow tetrahedral cage with large, keyhole-shaped pores in the shell (Fig. 3e,k; Supplementary Fig. 5b). Such assemblies are unprecedented (Supplementary Table 1).

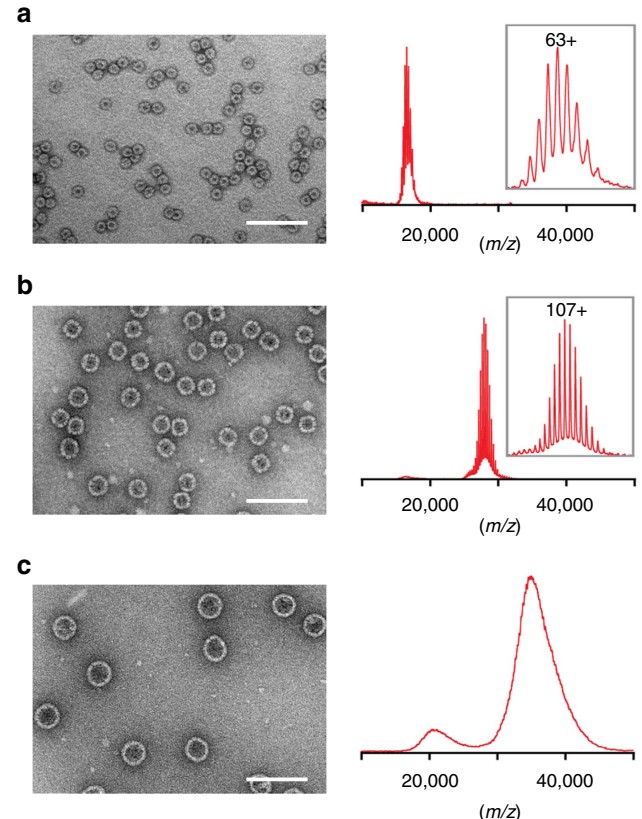

**Figure 2 | Size expansion of the AaLS variants.** (**a**–**c**) Negative-stain EM images and native mass spectra of AaLS-wt (**a**), AaLS-neg (**b**) and AaLS-13 (**c**). Scale bars, 100 nm. The most abundant charge states are annotated for AaLS-wt and AaLS-neg. The spectra clearly indicate that AaLS-wt is a 60-mer and AaLS-neg is a 180-mer. For AaLS-13, the oligomeric state was estimated to be between a 240-mer and a 420-mer extrapolating from the charging behaviour of the other AaLS variants.

We next turned to the larger AaLS-13 assembly. Despite some sample heterogeneity, an initial model for AaLS-13 could be reconstructed from the 2D class-averages of selected particles by imposing icosahedral constraints without using a structural reference (Supplementary Fig. 4). The 3D classification, followed by refinement using 9,900 particles, resulted in an EM map resolved to 5.2 Å resolution (Fig. 3c). The characteristic αβ-fold of the LS monomer units was clearly evident (Fig. 3i; Table 1). Like its evolutionary parent, AaLS-13 has a highly porous cage structure constructed exclusively from pentameric building blocks, but, in contrast to AaLS-neg, it adopts icosahedral symmetry (Fig. 3f,l; Supplementary Fig. 5c). The 39-nm wide, ∼6-MDa large sphere is assembled from 72 pentamers (360 subunits), twice as many as AaLS-neg. Its internal volume (15,600 nm$^3$) is 3.7 times larger than that of AaLS-neg (4,200 nm$^3$) and 58 times larger than that of AaLS-wt (270 nm$^3$), accounting for its enhanced loading capacity[22].

The AaLS-13 cage architecture can be best viewed as an extended dodecahedron, the dual counterpart of an icosahedron, with each face consisting of a central pentamer surrounded by five others (12 × 30-mer) (Supplementary Fig. 5d,e). Comparison of this structure with the icosahedral capsids of polyoma and papilloma viruses is instructive. The latter are also constructed from 72 pentameric building blocks, but the pentamers occupy both the pentavalent and hexavalent sites of a skewed $T = 7d$ icosahedral surface lattice[33,34] (Fig. 4b). Due to differences in

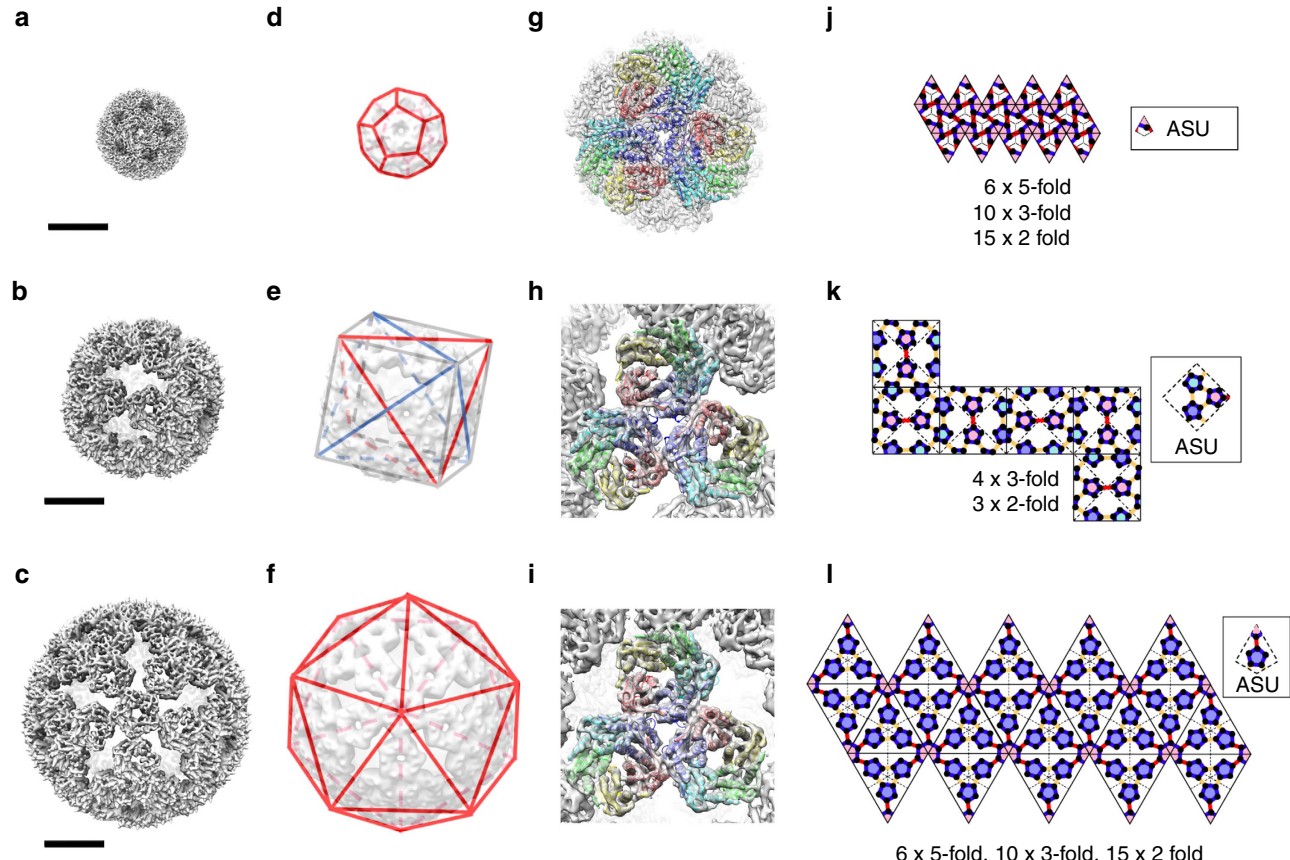

**Figure 3 | Cryo-EM structures of the AaLS variants.** (**a**–**c**) Cryo-EM density maps of AaLS-wt (**a**), AaLS-neg (**b**) and AaLS-13 (**c**) at 3.9, 5.4 and 5.2 Å resolution, respectively. Scale bars: 10 nm. (**d**–**f**) Dodecahedral, tetrahedral and icosahedral lattices matching the respective symmetries of AaLS-wt (**d**), AaLS-neg (**e**) and AaLS-13 (**f**). For AaLS-neg, two tetrahedral lattices (red and blue), embedded in a cube (grey), are depicted. (**g**–**i**) Close-up views of cryo-EM density maps of AaLS-wt (**g**), AaLS-neg (**h**) and AaLS-13 (**i**) around the three-fold rotational symmetry axes. Three pentamers from the refined models are shown as coloured ribbon diagrams. (**j**–**l**) 2D representation of the symmetry lattices for AaLS-wt (**j**), AaLS-neg (**k**) and AaLS-13 (**l**). Although the AaLS-neg structure is projected onto a flattened cube, note that it lacks the four-fold symmetry axes of a true cube because of the distinct pattern of pentamers on each face of the object and hence only possesses tetrahedral symmetry. The contact bars between two adjacent pentamers that have wt-like (pseudo) two-fold rotational symmetry are shown in red, while the other contact bars are shown in yellow. The asymmetric unit (ASU) for each structure is shown in the inset.

local environment, contacts between the capsomers at individual sites in the latter structures are not quasi-equivalent[19] but mediated instead by the flexible tails of the shell proteins[34]. In contrast, the AaLS-13 assembly is not skewed (Fig. 4a,c), and its porous surface avoids offending contacts that could arise when a pentamer is packed inside a pseudohexagonal hole. Such topology has never been observed in nature (Supplementary Table 2). It is, however, reminiscent of the aperiodic pentagonal packings considered by Kepler in 1619 (ref. 35) and later by Penrose[36].

**Stability**. Wild-type AaLS is a remarkably thermostable protein $(T_m = 120\,°C)$[14]. Dynamic light scattering (DLS) measurements confirm that the AaLS-wt cage remains intact beyond 90 °C. Despite their highly porous nature, the mutant AaLS-neg and AaLS-13 cages are also quite stable (Fig. 5a–d). DLS measurements suggest that they do not disassemble below 80 °C, and negative-stain EM verified that the cage structures even survive heating for 5 min at 90 °C (Fig. 5e–j). Nevertheless, nanoindentation experiments using atomic force microscopy showed that the AaLS-13 cage is mechanically soft with a 20-fold smaller Young's modulus than AaLS-wt (ref. 37).

**Discussion**

Natural protein cages are usually organized based on symmetries present in cubic point groups. For instance, 12 and 24 copies of a single polypeptide can assemble into tetrahedrally and octahedrally symmetric objects[2,38–40], respectively, whereas 60 polypeptides, or multiples thereof, form icosahedra[5,20], as exemplified by viral capsids[19,41]. With few exceptions[33,34,42–44], icosahedral structures contain 12 pentamers plus $10 \times (T - 1)$ hexamers, arranged to maintain quasi-equivalent inter-subunit interactions between individual polypeptides[19].

In the structures of the AaLS variants, the pentamer is the common and exclusive building block for all assemblies. While all pentamers in the wt cage are symmetrically equivalent, the AaLS-13 and AaLS-neg structures contain either two or three types of symmetrically non-equivalent pentamers, respectively (Fig. 3j–l; Supplementary Fig. 5). Superposition of AaLS-neg pentamers onto AaLS-wt pentamers revealed that the individual subunits of the engineered variant are slightly tilted outwards from the five-fold rotational symmetry axes (Cα RMSDs of 2.1–2.3 Å) (Supplementary Fig. 6d,f). This conformational change, which modifies the wedge-like shape of the pentamer (Fig. 6b), is likely the result of electrostatic repulsion between the anionic residues that were introduced on the lumenal surface (Fig. 6a,b and

**Table 1 | Refinement and validation statistics for the atomic models of AaLS-wt, AaLS-neg and AaLS-13.**

| Particle | AaLS-wt | AaLS-neg* | AaLS-13* |
|---|---|---|---|
| *Data collection* | | | |
| No. of particles | 3,268 | 26,769 | 9,900 |
| Applied symmetry | icosahedral | tetrahedral | icosahedral |
| Pixel size (Å) | 1.39 | 1.39 | 1.39 |
| Defocus range (μm) | −1.2 to −3.4 | −1.2 to −3.4 | −1.2 to −3.4 |
| Voltage (kV) | 300 | 300 | 300 |
| Electron dose (e$^-$/A$^2$) | 25 | 25 | 25 |
| | | | |
| *Reciprocal space data* | | | |
| Spacegroup | P1 | P1 | P1 |
| $a = b = c$ (Å) | 252.00 | 420.00 | 448.00 |
| $\alpha = \beta = \gamma$ (°) | 90.0 | 90.0 | 90.0 |
| | | | |
| *Refinement* | | | |
| Resolution range (Å) | 252.0-3.94 | 420.0-5.50 | 448.0-5.20 |
| Applied geometry weight (wxc) | 0.7 | 0.7 | 0.6 |
| No. reflections | 546,600 | 930,291 | 1,335,883 |
| R-factor | 0.250 | 0.215 | 0.248 |
| No. of protomers in ASU/particle | 1/60 | 15/180 | 6/360 |
| No. of residues | 9,240 | 28,980 | 57,960 |
| B-factor overall | 69.6 | 217.6 | 275.3 |
| R.m.s. deviations | | | |
| Bond lengths (Å) | 0.009 | 0.008 | 0.008 |
| Bond angles (°) | 1.00 | 1.03 | 1.02 |
| | | | |
| *Validation* | | | |
| Molprobity clashscore | 5.0 | 14.4 | 11.4 |
| Ramachandran plot | | | |
| Favoured (%) | 94.7 | 91.9 | 90.2 |
| Allowed (%) | 3.3 | 6.5 | 7.0 |
| Outliers (%) | 0.0 | 1.6 | 2.8 |

*To stabilize the refinement, all atoms were used; however the final model does not include side chains because they are not resolved.

Supplementary Movie 1). Repositioning of the α4 helix (residues 120–131) (Supplementary Fig. 6a–c), which contains two of the designed mutations, further modifies the shape of the pentamer wedge. This secondary structural element appears to be relatively flexible, judging by its weak density and high B values, allowing it to adopt different conformations depending on the interaction partner. These effects are even more pronounced in AaLS-13 (Cα RMSDs of 2.8 Å) (Supplementary Fig. 6e,f), consistent with the increased number of anionic residues incorporated by directed evolution (Fig. 6a,b; Supplementary Movie 2). The A118D mutation, in particular, induces a substantial displacement of the α4 helix to avoid steric and electrostatic clashes.

How does the small conformational change in each pentamer trigger the observed global geometrical rearrangements? The modified pentamer wedges (Fig. 6b) would cause steric clashes if the cage maintained its original radius of curvature. To avoid such clashes, the angle between the local five-fold rotational symmetry axes of adjacent pentamers decreases, leading to formation of larger cages (Fig. 6c). The respective angles for AaLS-neg and AaLS-13 are in the range of 21.4–39.5° and 20.8–28.5°, significantly reduced from the 63.4° angle seen in AaLS-wt (Supplementary Fig. 7; Supplementary Movies 3,4).

The different angles found for the variants dictate the global structures and symmetries of the expanded cages. In the wt structure, two distinct inter-pentamer interfaces exist near the two- and three-fold rotational symmetry axes (Supplementary Fig. 8a–d). The three-fold interfaces located on the lumenal side of the cage are substantially reduced in size on increasing cage curvature (Supplementary Fig. 8e–g). In contrast, the two-fold interfaces near the outer surface of the cage are well maintained. Morph movies between AaLS-wt and the mutants show that these

interfaces function as 'hinges' between two pentamer wedges (Supplementary Movie 5). The position of one protomer is rotated ∼20° relative to its neighbour, changing the curvature of the cage while still maintaining contacts in the hinge region (Supplementary Fig. 9). Polymorphism in the cowpea chlorotic mosaic virus capsid, which can adopt different quasi-equivalent states, has been ascribed to a similar inter-subunit hinge movement[45].

A remarkable feature of the negatively supercharged AaLS assemblies is that roughly 30% of the protomers (48/180 for AaLS-neg and 120/360 for AaLS-13) lack inter-pentamer interactions, leaving large pores in the shell wall. AaLS-neg and AaLS-13 possess 12 and 30 such openings, respectively, yet maintain high thermal stability. The solvent-exposed surfaces that define the edges of the keyhole-shaped slots do not show obvious architectural differences compared to their buried counterparts (Fig. 7a,b). At their centre, the openings are ∼4 nm wide and thus significantly larger than the channel located at the centre of each pentamer (<1 nm), allowing free passage across the shell for molecules with dimensions up to 4 nm. In combination with their supercharged interiors, this structural feature allows the AaLS mutants to function as 'protein sponges' that absorb complementarily charged guest molecules at rates approaching the diffusion limit[25] (Fig. 7c,d). Since AaLS-wt can also adopt expanded cage structures under certain conditions[28], similar openings may mediate the release of enzymatically produced riboflavin, which is too large to pass through the <1 nm channels at the centre of the AaLS pentamers[20].

In nature, a single protein fold can be evolved to form multiple oligomeric states with different symmetries. The current study shows that such transformations can also be discovered by design

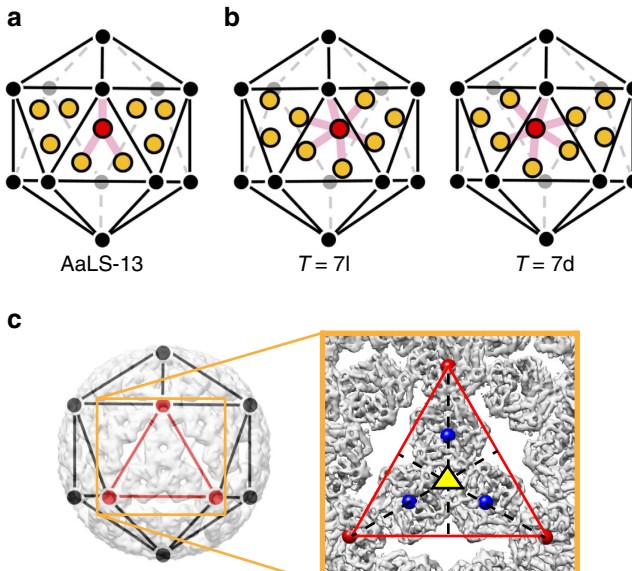

**Figure 4 | AaLS-13 capsomer geometry. (a,b)** Positions of capsomers in AaLS-13 (**a**) compared with their locations in skewed $T = 7l$ and $T = 7d$ viral capsids (**b**). The twelve pentavalent vertices of the icosahedral lattice are shown as black and grey circles. For AaLS-13, pentameric capsomers, projected onto three triangular facets of the lattice, are shown as ochre or red circles; each interacts directly with only three other capsomers, as illustrated by the pink lines from the red unit. For the $T = 7l$ and $7d$ structures, the corresponding hexavalent sites are projected onto the same facets. In this case, the capsomer units have six neighbours each (shown by the pink lines from the red capsomer). (**c**) Overlay of an icosahedral lattice on the AaLS-13 structure. The cryo-EM density map viewed along one of the three-fold symmetry axes (yellow triangle) is shown magnified in the box on the right. Centroids of pentamers at five-fold symmetry axes are shown as red spheres. Centroids of pentamers projected onto an equilateral triangular facet of the icosahedral lattice are shown as blue spheres.

and evolution in the laboratory. The novelty of the cage architectures studied here may serve as inspiration for ongoing efforts to create novel oligomeric protein assemblies. Completely artificial protein cages have been generated based on symmetries present in cubic point groups by designing building blocks with sterically complementary interfaces, either by site-directed mutagenesis[46–49] or by using covalent linkages[50,51]. Formation of tetrahedrally and icosahedrally symmetric particles entirely from pentamers demonstrates that non-equivalent arrangements of individual building blocks may also be considered for the construction of isometric shells. The dramatic morphological changes observed in the supercharged AaLS variants additionally highlight the importance of mutations distant from the capsomer interface for determining overall cage architecture. By coupling such structural rearrangements in the individual building blocks to an external stimulus, such as a change in pH or ionic strength, it may even be possible to design structures that undergo controlled (dis)assembly.

The expanded repertoire of hierarchically ordered nanostructures formed by supercharged AaLS variants offers novel insights into the principles underlying the construction, function and evolution of protein assemblies. These highly plastic protein cages represent a versatile platform for creating tailored structures for a plethora of practical applications in nano- and medical science.

## Methods

**Preparation of AaLS variants.** Plasmids encoding AaLS-wt, AaLS-neg, and AaLS-13 proteins were previously described[21,22]. Each protein was overexpressed with a

C-terminal His$_6$-tag in *E. coli* BL21-gold (DE3) cells and purified by Ni$^{2+}$-NTA affinity chromatography (Qiagen) followed by size-exclusion chromatography using a HiPrep 16/60 Sephacryl S-400 HR column (GE Healthcare Life Sciences) with 20 mM Tris–HCl buffer (pH 7.8) containing 0.2 M NaCl and 1 mM EDTA according to published protocols[29].

For native MS studies, AaLS-neg and AaLS-13 lacking a His$_6$-tag were prepared. The respective genes were amplified from the plasmids encoding the corresponding His$_6$-tagged variants by PCR with primers AaLS_fw (5′-AGA TAT ACA TAT GGA AAT CTA CGA AGG) and AaLS_noHis_rv (5′-GGT GGT GCT CGA GTC ATT ATT ATC GGA GAG ACT TGA ATA AGT TTG C). After digestion with NdeI and XhoI, each fragment was ligated into the original plasmid, which had been digested with the same restriction enzymes, to give pMG211_AaLS-neg_noHis and pMG211_AaLS-13_noHis. The proteins lacking the His$_6$-tag were overexpressed in *E. coli* BL21-gold (DE3) cells, and the cell pellets from 0.4 l culture were resuspended in 15 ml of 50 mM sodium phosphate buffer (pH 8.0) containing 0.3 M NaCl, 1 mg ml$^{-1}$ of lysozyme, 1 mg ml$^{-1}$ of polymixin, EDTA-free protease inhibitor (complete protease inhibitor cocktail tablets, Roche), and a spatula tip of both DNase I and RNase A (Sigma Aldrich). After incubating at room temperature for one hour, the cells were disrupted by sonication and cleared by centrifugation. The crude lysates containing AaLS-neg were incubated at 85 °C for 20 min. For AaLS-13, 0.75 volumes of saturated aqueous ammonium sulphate were added to the lysates at room temperature. Following removal of the precipitate by centrifugation, the clear supernatants were buffer exchanged into 50 mM sodium phosphate buffer (pH 8.0) containing 0.2 M NaCl. The treated lysates were subjected to anion exchange chromatography using a MonoQ 10/100 GL column (GE Healthcare Life Sciences) preequilibrated with 50 mM sodium phosphate buffer (pH 8.0) containing 0.2 M NaCl and eluted by a linear gradient generated by increasing the NaCl concentration to 1 M. The protein fractions were buffer exchanged into 50 mM Tris–HCl buffer (pH 7.8) containing 1 M NaCl and concentrated until the monomer concentration exceeded 1 mM. After incubation over 7 days at room temperature, the protein samples were further purified by size-exclusion chromatography using a HiPrep 16/60 Sephacryl S-400 HR column with 50 mM Tris–HCl buffer (pH 8.0) containing 1 M NaCl as mobile phase. AaLS-13 samples were supplemented with 3 mM DTT to prevent formation of disulphide bonds.

**Native MS.** Purified AaLS variants were prepared for native MS by exchanging the buffer to 150 mM ammonium acetate (Sigma Aldrich), pH 7.8 through seven consecutive dilution and concentration steps using a 10-kDa molecular weight cutoff filter (Millipore). For native MS analysis, aliquots of the samples were diluted to a monomer concentration of 5 µM in 150 mM ammonium acetate, 25 mM triethylammonium acetate (Fluka), pH 7.8. For tandem MS experiments, aliquots of the samples were diluted in 150 mM ammonium acetate pH 7.8. For dissociation of AaLS-neg into smaller capsomers, samples were prepared in 10 mM ammonium acetate, pH 7.8. Experiments were performed on a quadrupole time-of-flight (qTOF) MS (MS Vision, Waters) modified for the analysis of high molecular weight ions[52]. AaLS-WT, AaLS-neg and AaLS-13 were introduced into the gas phase through the nano-electrospray interface using gold-coated borosilicate capillaries prepared in-house. Capillary voltages were set at ∼1,400 V in positive ion mode. Pressure in the xenon-filled collision cell was $1.8 \times 10^{-2}$ mbar. For native MS experiments, typical settings were 100 V sample cone and 150–250 V collision voltage. For tandem MS experiments, typical settings were 50 V sample cone voltage and a collision voltage ramping from 50 to 300 V. The instrument was calibrated using caesium iodide clusters up to 30,000 *m/z*. Spectra were analysed using Masslynx 4.1 software (Waters). Masses were determined manually by minimization of the error over the charge state envelope from different charge state assignments.

**Cryo-EM data collection.** Purified samples (1.5–2.0 µl each) of AaLS-wt, AaLS-neg, and AaLS-13 cages (monomer concentration of 80, 80 and 300 µM, respectively) in 20 mM Tris–HCl buffer (pH 7.8) containing 0.2 M NaCl and 5 mM ethylenediaminetetraacetic acid (EDTA) were applied to holey carbon grids R2/2 (Quantifoil), which had been glow-discharged for 90 s using an Emitech K100X glow discharge system. The grids were blotted for 4 s twice at ambient temperature and 100% humidity and then plunged into liquid ethane using a Vitrobot mark one unit (FEI Company). Data were recorded semi-automatically using the EPU software on a Titan Krios transmission electron microscope (TEM) operated at 300 kV and equipped with a Falcon II direct electron detector (FEI Company). Images were acquired at −1.2 to −3.4 µm defocus at 100,720-fold magnification, resulting in a pixel size of 1.39 Å on the object scale. Beam-induced motion correction was further performed for images of AaLS-wt and AaLS-13. Specifically, the images were recorded as seven separate frames, comprising a total exposure of 25 electrons Å$^{-2}$, which were subsequently aligned and summed using DOSEFGPU DRIFTCORR to obtain a final image[53].

**Cryo-EM data processing.** Estimation of the contrast transfer function was performed using CTFFIND3 (ref. 54). Micrographs exhibiting poor power spectra based on the extent and regularity of the Thon rings were rejected. Particles were picked using EMAN 2.1 (ref. 55) or RELION 1.4 (ref. 56). 2D reference-free

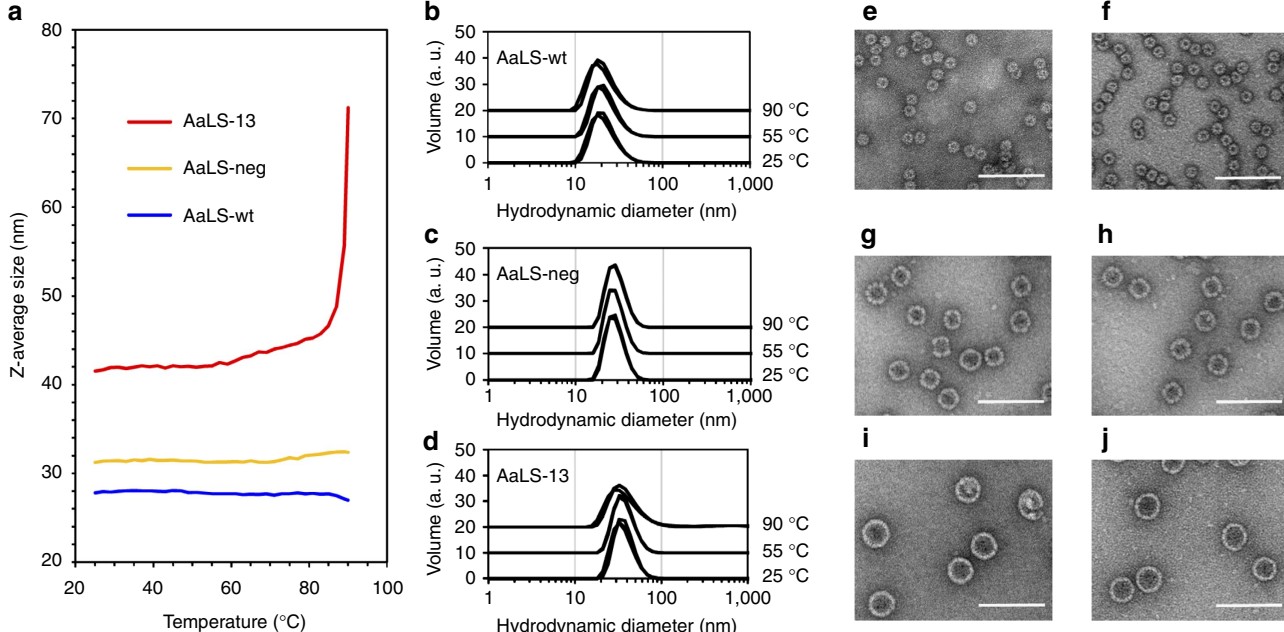

**Figure 5 | Thermal stability of the AaLS variants. (a)** DLS measurements with AaLS-wt (blue), AaLS-neg (ochre) and AaLS-13 (red). Z-Averages[68] of the particles are plotted against the measured temperature (25–90 °C). The Z-averages of AaLS-neg (∼32 nm) and AaLS-13 (∼42 nm) correspond to the expected hydrodynamic diameters expected for 29- and 39-nm wide cages, respectively. However, the Z-average (∼28 nm) for AaLS-wt was significantly larger than the expected hydrodynamic diameter of a 16-nm wide cage, suggesting that a fraction of the particles aggregate in solution. **(b–d)** Volume weighted distribution of the hydrodynamic diameters determined for AaLS-wt (**b**), AaLS-neg (**c**) and AaLS-13 (**d**) at 25, 55 and 90 °C. The individual histograms are offset by 10 arbitrary volume units (a.u.) for clarity. Three independent measurements were performed and plotted for each condition. **(e–j)** Negative-stain EM images of AaLS-wt (**e,f**), AaLS-neg (**g,h**) and AaLS-wt (**i,j**). Each sample was deposited on the EM grid without heating (**e,g,i**) or immediately after heating for 5 min at 90 °C (**f,h,j**). Scale bars, 100 nm.

alignments of selected particles in 50 classes for AaLS-wt and AaLS-13 and 80 classes for AaLS-neg were performed by RELION 1.4. Particles that did not yield high-resolution class-averages were excluded from further refinement (Supplementary Figs 2–4 for more detailed workflows). Images of the selected 2D class-averages were used to reconstruct initial low-resolution models by imposing icosahedral symmetry for AaLS-wt and AaLS-13 or tetrahedral symmetry for AaLS-neg with the program e2initialmodel.py in EMAN 2.1. 3D classification was then performed with RELION 1.4 using the low-pass filtered initial models to 50 Å as references. Further 3D classification and refinement were performed using full-sized images (1.39 Å per pixel) and masks excluding the cage cavities.

**Cryo-EM structural analysis.** The cryo-EM maps and the molecular structures were analysed using UCSF Chimera 1.11 (ref. 57) and PyMOL (Version 1.7, Schrödinger LLC). Local resolutions of the EM maps were estimated using the ResMap program[58]. Electrostatic surface potentials were calculated for the crystal structure (PDB: 1HQK)[14], for AaLS-wt, and mutation imposed crystal structure, for AaLS-neg and AaLS-13, at pH 8 and an ionic strength of 150 mM using PDB2PQR (refs 59,60) and APBS (Adaptive Poisson-Boltzmann Solver)[61] tools through UCSF Chimera. The default settings were used for the calculations except for the pH and ionic strength parameters. The graphics used for the figures and movies were generated using UCSF Chimera or PyMOL.

**Atomic model building and refinement.** For generation of a coordinate model that includes the entire AaLS-wt particle, one protomer of the corresponding crystal structure (PDB: 1HQK)[14] was adjusted to the EM density using the program O (refs 62,63) and symmetry expanded using the icosahedral operators from RELION 1.4 (ref. 56). The resulting model was subjected to phase-restrained reciprocal space refinement in PHENIX (ref. 64) (Table 1) against amplitudes and phases back-calculated from the 3.9 Å EM map, as described in ref. 65. During initial steps of coordinate and individual B-factor refinement, secondary structure and torsional non-crystallographic symmetry restraints were applied. The weighting of the coordinate geometry versus the map term was set to a value that resulted in a good overall model geometry during refinement (Table 1). For the final refinement round, one monomer was re-expanded, and the atomic model of the particle was minimized against the EM map imposing a strict 60-fold non-crystallographic symmetry. In the $F_{obs}-F_{calc}$ difference Fourier map, additional density can be identified close to the active site, which likely represents a reaction intermediate. However, as none of the ligands available from crystal structures solved in complex with substrate analogues[66,67] convincingly explains the

difference density and the resolution of the EM map does not allow modelling of a ligand at atomic detail *de novo*, we left this area of the map uninterpreted.

For the AaLS-neg and AaLS-13 mutants, the affected residues were exchanged in one of the 1HQK protomers, which then was expanded to 15 (AaLS-neg) or 6 (AaLS-13) positions to generate the asymmetric unit (ASU) of the particles. Areas of the ASU models located outside of the EM-density were manually readjusted in each monomer, which included a stretch around helix α4 (residues 117–134) and a neighbouring loop (residues 82–90). A few residues were added to the C-terminal α5 helix, which extends further towards the periphery of the particles. After tetrahedral 12-fold (AaLS-neg) or icosahedral 60-fold (AaLS-13) symmetry expansion, the full model was refined using a similar strategy as outlined above for AaLS-wt, except for B-factor refinement, which was performed in a 'two groups per residue' mode (Table 1). Because of the limited resolution in the ∼5 Å range, in which only large bulky residues are visualized, the refined model was truncated to the backbone.

**DLS.** AaLS-wt, AaLS-neg and AaLS-13 purified by SEC were diluted in 50 mM Tris–HCl buffer (pH 7.8) containing 0.2 M NaCl and 5 mM EDTA to the final protein monomer concentration of 0.5 mg ml⁻¹ (corresponding to 30 μM of AaLS monomer) before DLS analysis. The protein samples were measured at 25–90 °C with a 2 °C interval using a Zetasizer Nano ZS instrument (Malvern Instruments). Data analysis was performed using the Zetasizer software 7.11 (Malvern Instruments).

**Negative-stain EM.** AaLS-wt, AaLS-neg, and AaLS-13 purified by SEC were diluted in 20 mM Tris–HCl buffer (pH 7.8) containing 0.2 M NaCl and 1 mM EDTA to a final protein monomer concentration of 0.06–0.08 mg ml⁻¹ (corresponding to 2–3 μM of AaLS monomer). The resulting samples were either left at room temperature or heated at 90 °C for 5 min in microcentrifuge tubes using Thermomixer Compact (Eppendorf). Immediately after heating or without heating, a 4-μl aliquot of each sample was deposited onto glow discharged copper grids with carbon support films (01814-F, Ted Pella). The grids were washed with the corresponding buffers and stained with 2% uranyl acetate before TEM analysis on a Morgagni 268 (FEI Company) microscope operated at 100 kV.

**Data availability.** The data that support the findings of this study are available from the corresponding author on reasonable request. The cryo-EM density maps have been deposited in the EM Data Bank with the accession codes EMD-3538 (AaLS-wt), EMD-3543 (AaLS-neg) and EMD-3544 (AaLS-13). Atomic coordinates

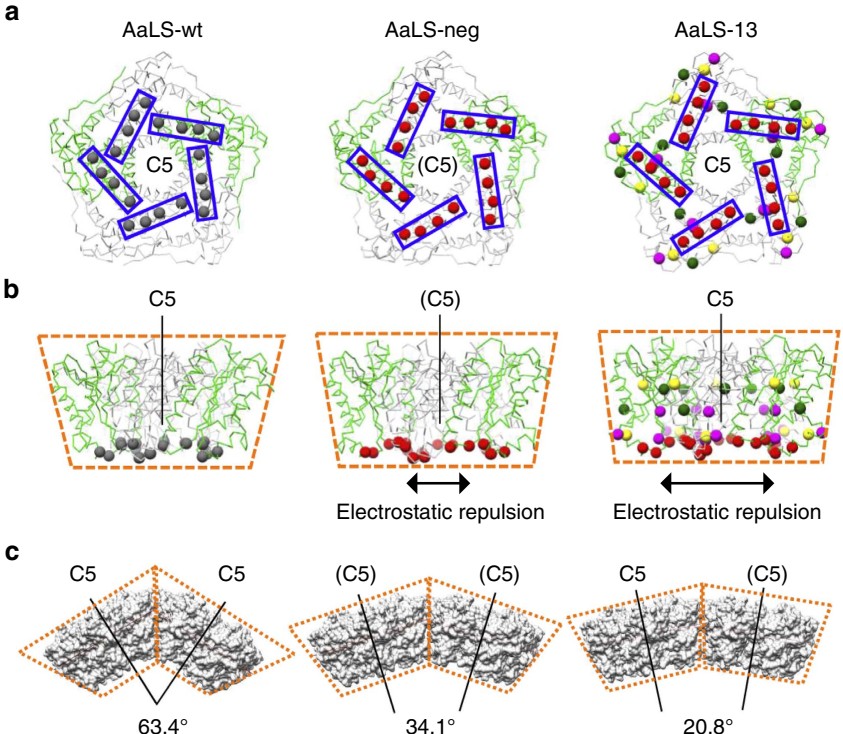

**Figure 6 | Geometrical rearrangements triggered by tilted monomers.** (**a,b**) Representative pentamers of AaLS-wt, AaLS-neg and AaLS-13. Two of the five monomers for each structure are shown in green and the rest in grey. Views around the (pseudo) five-fold rotational symmetry axes (**a**) and the corresponding side views parallel to the axes (**b**) are shown. Residues 83, 86, 120 and 123, which were mutated to glutamates by design, are shown as red spheres and boxed. For AaLS-13, residues 28, 52, 112, 115, 118, 127 and 131, which were mutated during directed evolution, are shown as coloured spheres. Red and magenta spheres indicate the negatively charged residues that were introduced. The wedge-like shapes of the pentamers are schematically highlighted by dotted lines. (**c**) Representative orientation of two adjacent pentamers in AaLS-wt, AaLS-neg and AaLS-13, respectively. Angles between the adjacent local (pseudo) five-fold axes are shown.

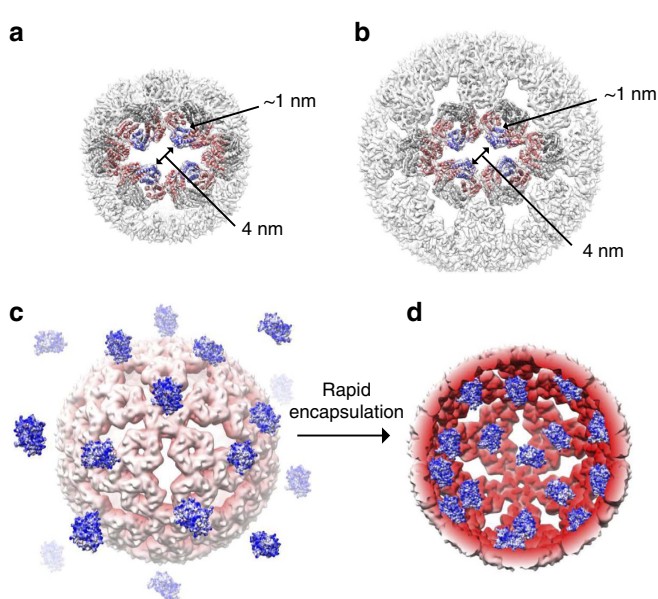

**Figure 7 | Diffusion-limited protein cargo loading.** (**a,b**) The keyhole-shaped openings of AaLS-neg (**a**) and AaLS-13 (**b**). Each slot is surrounded by 16 protomers, 4 of which (shown in blue) lack the inter-pentamer interactions. (**c,d**) Scheme illustrating spontaneous encapsulation of supercharged GFP molecules by AaLS-13. A half-sliced AaLS-13 cage is depicted (**d**) to show association of the guest with the cage interior.

of the corresponding models have been deposited in the Protein Data Bank with the accession codes 5MPP (AaLS-wt), 5MQ3 (AaLS-neg) and 5MQ7 (AaLS-13).

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

## Acknowledgements

This work was supported by the European Research Council (Advanced ERC Grant ERC-dG-2012-321295 to D.H.), JSPS Postdoctoral Fellowships for Research Abroad (E.S.), the Stipendienfonds der Schweizerischen Chemischen Industrie (R.Z.), and the Netherlands Organization for Scientific Research (Proteins@Work project 184.032.201) and Fundamenteel Onderzoek der Materie (Projectruimte grant 12PR3303-2) to M.v.d.W. and A.J.R.H. The cryo-EM data were collected at the Scientific Center for Optical and Electron Microscopy (ScopeM, ETH Zürich) with excellent support from P. Tittmann. We thank R. Frey for his help with protein purification and T. Beck and A. Bach for technical assistance with the computational settings.

## Author contributions

All authors contributed to the experimental design. E.S. performed the cryo-EM experiments. E.S., D.B., M.L., N.B. and D.H. analysed the cryo-EM data. M.v.d.W. and

A.J.R.H. carried out the native MS studies on samples prepared by R.Z. The manuscript was written by E.S. and D.H. and discussed and edited by the other authors.

## Additional information

**Competing interests:** The authors declare no competing financial interests.

