## [Peer Review File · Nature Communications]

Reviewers' comments:

Reviewer #1 (Remarks to the Author):

This paper describes the EM characterization of a set of large protein cages based on the lumazine synthase system, with different cage variants arising from introduction of interior charges (for encapsulating other cargo) and from further evolved mutations. Overall the paper is remarkable. The results show convincingly that the larger-than $T=1$ cages are assembled *not* based on well-established principles of quasi-equivalence, but rather on surprising schemes in which certain edge interactions are preserved, while gaps and unsatisfied edge contacts are allowed. Designing protein cages and other assemblies is currently in an exciting phase of scientific development, and the surprising finding of this paper provide much food for thought. The observations are truly novel and should be as interesting to current researchers as they would have been to Kepler 400 years ago! I tried to find places where substantial improvements could be made, but came up only with one or two points.

The pentamer arrangements in the AaIS-neg case are the most complex. I feel that the diagrammatic choice in figure 3k could probably be done better in a different way. A chief difficulty is that the flattened scheme is drawn on a cube, whereas the assembly only has tetrahedral (and not cubic/octahedral) symmetry. I think this case would be much clearer if it was redrawn instead on a flattened tetrahedron. Then one might have a better chance of understanding how the 9 pentamers are arranged (according to 3-fold symmetry) on each of the 4 triangular faces.

The comment about diffusion limited absorption on line 46 may not convey much meaning to some readers. It could be more informative to say that the encapsulation process "is rapid enough to be limited mainly by the rate of cargo diffusion."

It may not be helpful for the current presentation, but it strikes me that the scheme observed in the AaIS-13 case is similar to schemes considered as early as Kepler (1619) while trying to classify aperiodic tilings of the plane using pentamers and stars and other shapes. See attached figure.

-Todd Yeates

Reviewer #2 (Remarks to the Author):

The ms describes the structural characterization of spherical shells of lumazine synthase. The authors have previously mutated the protein to add negative charges followed by directed evolution. They have previously shown that the protein produces two species of expanded shells (by negative stain microscopy) and that the shells can package highly charged proteins.

In the present ms, the authors use mass spectrometry to show that the mutant shells have a specific molecular mass and therefore subunit number, and they have studied the shell structure by cryomicroscopy at sub-nanometer resolution.

A cryoEM map (3.9 Å) shows a wild-type $T=1$ shell similar to the known crystal structure, an assembly of pentameric subunits. Maps of the mutant shells (5.4 Å, 5.2 Å) both show surprising and novel architectures in that they are composed entirely of pentamer subunits which combine with either tetrahedral or icosahedral symmetry, and these packing arrangements require large fenestrations. The fenestrations allow access to the interior and likely provide a route for proteins to enter the shell interior. The results extend the repertoire of architectures used to build protein shells, first considered in quasi-equivalence theory of virus capsid structure.

These results are of interest to an audience of structural biologists studying self-assembly, protein engineers, but also to a wider audience interested in materials. The paper is clearly written, and

the analysis of the symmetry is thorough and informative. The figures, tables, and movies are informative.

The main limitation of the study is that the resolutions of the maps are probably inadequate for a more detailed molecular understanding of how the mutations alter the geometry, although the overall geometric changes are well-characterized. A role of electrostatics in determining the architecture is plausible.

Major point:

1. The authors suggest that the alpha 4 helix is flexibly repositioned in forming different inter-pentamer interactions in building shells of different sizes and curvature. I feel it is important that the authors show experimental cryoEM map density for this feature in wild-type and mutants to substantiate this important observation.

Minor points:

1. P. 6 Sentence fragment "Z-Averages." must be a typo.
2. p. 9 "Solvent-exposed surfaces that define the edges of the keyhole-shaped slots do not show structural perturbations compared to their buried counterparts." This is an interesting observation, but I assume there will be differences at higher resolution.
3. I don't think Figure 7c adds to our understanding of the system so think it could be removed.
4. I don't know that the cages are really "scalable" or scalable on demand, as implied by the title.

Reviewer #3 (Remarks to the Author):

This is a study of large complexes of the capsid-forming enzyme AaLS employing native mass spectrometry and cryo-electron microscopy to characterize the cage-like structures of these macroassemblies. Three mutants are investigated: the wild type AaLS-wt; an engineered mutant AaLS-neg with four mutations to negatively charged glutamates that project into the lumen of the capsid; and another variant, AaLS-13, obtained by directed evolution and featuring an additional seven mutations which further increase the negative charge in the container. It is found that AaLS-wt forms a 60-mer complex in the overall shape of an icosahedron, that AaLS-neg forms a 180-mer with tetrahedral symmetry, and that AaLS-13 forms a 360-mer in the shape of a dodecahedron. The cage structures are stable up to 80 degrees Celsius. It is explained how an increase of negatively charged residues on one side of the monomer structure leads through Coulomb repulsion to a change in the monomer structure which in turn leads to a decrease in the curvature of assembled oligomers which finally leads to larger oligomer assemblies.

The paper is generally well written, the research is outstanding, and the topic is very interesting to a broad audience of chemists. I recommend publication of this article after a few minor changes.

Detailed comments:

1.) Abstract, line 17: "60-subunit dodecahedron". For the general audience not intimately familiar with virus and similar structures, it is odd to make a connection between 60 units and the shape of a dodecahedron. Later, in the text of the paper it becomes obvious how this geometry is assembled but in the abstract it is confusing. For the expert this is not an issue, of course, but I would like to bring this point to the attention of the authors and perhaps there is a simple way to alleviate the mismatch between the numbers 12 (shape) and 60 (units).

2.) The first paragraph of the "Single-particle reconstruction" section, lines 76-84, can only be understood by experts familiar with the procedure. The authors should make an attempt to rephrase this paragraph to make it more readable for a general audience without expanding too much, though. An example of a very successfully explained complex procedure is the brief explanation of "directed evolution" in one sentence (lines 41-43).

3.) Lines 123-124: There is a fraction of an incomplete sentence, perhaps a leftover of a draft version. Also, it should say "using atomic force microscopy" or "using an atomic force microscope" instead of "using atomic force microscope".

4.) Line 140: "... the wedge angle of the pentamer...". I was not immediately able to understand which angle is referred to. I had to study all figures carefully and read the text in its entirety before I could make sense of this phrase. (It's an angle perpendicular to the plane of the pentamer pentagon. Initially, I tried to see the pentagon shape itself as a wedge which simply did not make any sense in the following discussion.) I think a reference to figure 6c would be appropriate here: "... the wedge angle of the pentamer (Fig. 6c)...".

5.) Line 164: "The position of one protomer is rotated $\sim 20^\circ$ relative to its neighbor, increasing the curvature of the cage while still maintaining contacts in the hinge region (Supplementary Fig. 9)." Increasing the curvature by going from which mutant to which other mutant? A better way would be to replace "increasing" by "changing": "The position of one protomer is rotated $\sim 20^\circ$ relative to its neighbor, thereby changing the curvature of the cage while still maintaining contacts in the hinge region (Supplementary Fig. 9)."

6.) Line 283: "... one protomer... was adjusted to the EM density using O...". What is "O"?

Reviewer #1 (Remarks to the Author):

This paper describes the EM characterization of a set of large protein cages based on the lumazine synthase system, with different cage variants arising from introduction of interior charges (for encapsulating other cargo) and from further evolved mutations. Overall the paper is remarkable. The results show convincingly that the larger-than T=1 cages are assembled *not* based on well-established principles of quasi-equivalence, but rather on surprising schemes in which certain edge interactions are preserved, while gaps and unsatisfied edge contacts are allowed. Designing protein cages and other assemblies is currently in an exciting phase of scientific development, and the surprising finding of this paper provide much food for thought. The observations are truly novel and should be as interesting to current researchers as they would have been to Kepler 400 years ago! I tried to find places where substantial improvements could be made, but came up only with one or two points.

We are grateful for the Reviewer's enthusiastic support.

The pentamer arrangements in the AaLS-neg case are the most complex. I feel that the diagrammatic choice in figure 3k could probably be done better in a different way. A chief difficulty is that the flattened scheme is drawn on a cube, whereas the assembly only has tetrahedral (and not cubic/octahedral) symmetry. I think this case would be much clearer if it was redrawn instead on a flattened tetrahedron. Then one might have a better chance of understanding how the 9 pentamers are arranged (according to 3-fold symmetry) on each of the 4 triangular faces.

We appreciate the reviewer's comments regarding the flattened representation of the AaLS-neg structure in Fig. 3k. Although the structure possesses tetrahedral symmetry, the physical object does not resemble a tetrahedron but instead looks more like a rounded cube. It nevertheless lacks cubic/octahedral symmetry because of the different pentamer patterns on each face of the "cube". As a result, each face of this "cube" possesses a 2-fold symmetry axis rather than the 4-fold symmetry axis expected for a true cube. The reduction of octahedral to tetrahedral symmetry is illustrated in Fig. 3e, which shows that the cube (gray) and the embedded tetrahedra (red and blue) share the same 3-fold symmetry axes, whereas the 4-fold symmetry axes of the cube correspond to the 2-fold symmetry axes of the tetrahedra. Since other readers are likely to have the same concerns about Fig. 3k as the referee, we added the following sentence to the legend: "Although the AaLS-neg structure is projected onto a flattened cube, note that it lacks the 4-fold symmetry axes of a true cube because of the distinct pattern of pentamers on each face of the object and hence only possesses tetrahedral symmetry."

As shown in Supplementary Fig. 5b, when the flattened cube shown in Fig. 3k is folded, an object is obtained that resembles the experimental 3-D structure of AaLS-neg quite well. In contrast, if we redraw the AaLS-neg structures on a flattened tetrahedron as suggested by the reviewer (see below) and then fold it, this is not the case. This representation effectively illustrates the tetrahedral symmetry of AaLS-neg as well as the interactions between the symmetrically unrelated pentamers, but as noted above, the actual experimental structure is not a tetrahedron. As a consequence, we believe that the original Fig. 3k provides a better depiction of the true AaLS-neg structure than the alternative proposed by the referee.

The comment about diffusion limited absorption on line 46 may not convey much meaning to some readers. It could be more informative to say that the encapsulation process "is rapid enough to be limited mainly by the rate of cargo diffusion."

Modified as suggested.

It may not be helpful for the current presentation, but it strikes me that the scheme observed in the AaLS-13 case is similar to schemes considered as early as Kepler (1619) while trying to classify aperiodic tilings of the plane using pentamers and stars and other shapes. See attached figure.

-Todd Yeates

The link between our structures and Kepler's considerations is both interesting and worth mentioning in the main text. We therefore added the sentence "It is, however, reminiscent of the aperiodic pentagonal packings considered by Kepler in 1619 (ref. 35,36) and later by Penrose³⁷" to the end of the discussion of the AaLS-13 structure on page 6. We also thank the reviewer by name in the acknowledgments for alerting us to this precedent.

Reviewer #2 (Remarks to the Author):

The ms describes the structural characterization of spherical shells of lumazine synthase. The authors have previously mutated the protein to add negative charges followed by directed evolution. They have previously shown that the protein produces two species of expanded shells (by negative stain microscopy) and that the shells can package highly charged proteins.

In the present ms, the authors use mass spectrometry to show that the mutant shells have a specific molecular mass and therefore subunit number, and they have studied the shell structure by cryomicroscopy at sub-nanometer resolution.

A cryoEM map (3.9 Å) shows a wild-type T=1 shell similar to the known crystal structure, an assembly of pentameric subunits. Maps of the mutant shells (5.4 Å, 5.2 Å) both show surprising and novel architectures in that they are composed entirely of pentamer subunits which combine with either tetrahedral or icosahedral symmetry, and these packing arrangements require large fenestrations. The fenestrations allow access to the interior and likely provide a route for proteins to enter the shell interior. The results extend the repertoire of architectures used to build protein shells, first considered in quasi-equivalence theory of virus capsid structure.

These results are of interest to an audience of structural biologists studying self-assembly, protein engineers, but also to a wider audience interested in materials. The paper is clearly written, and the analysis of the symmetry is thorough and informative. The figures, tables, and movies are informative. The main limitation of the study is that the resolutions of the maps are probably inadequate for a more detailed molecular understanding of how the mutations alter the geometry, although the overall geometric changes are well-characterized. A role of electrostatics in determining the architecture is plausible.

We appreciate the Reviewer's constructive comments and suggestions.

Major point:

1. The authors suggest that the alpha 4 helix is flexibly repositioned in forming different inter-pentamer interactions in building shells of different sizes and curvature. I feel it is important that the authors show experimental cryoEM map density for this feature in wild-type and mutants to substantiate this important observation.

Although the density for the alpha 4 helix is comparatively weak in the cryo-EM maps of the engineered variants, initial fitting using the AaLS-wt monomer clearly showed that the original cryoEM map density corresponding to this structural element in the variants did not match with the

model. As recommended by the reviewer, we now provide close-up views of the experimental density for this feature together with the modified model of AaLS-neg in Supplementary Figure 6b.

Minor points:

1. P. 6 Sentence fragment “Z-Averages.” must be a typo.

Corrected.

2. p. 9 “Solvent-exposed surfaces that define the edges of the keyhole-shaped slots do not show structural perturbations compared to their buried counterparts.” This is an interesting observation, but I assume there will be differences at higher resolution.

The resolution of our models is only ~ 5 Å, so the reviewer is undoubtedly correct. We therefore replaced “do not show structural perturbations” with “do not show obvious architectural differences”.

3. I don’t think Figure 7c adds to our understanding of the system so think it could be removed.

Fig. 7c illustrates how the engineered cages function as “protein sponges”, which we believe to be important for a general audience. The graphic rationalizes the diffusion-limited encapsulation of supercharged proteins like GFP(+36) in pictorial fashion, showing that the keyhole-shaped openings in the AaLS-13 shell are large enough to provide direct access to the cage interior. It also shows that the cargo molecules likely pack against the luminal walls of the container. For these reasons, we strongly feel that this illustration should be retained.

4. I don’t know that the cages are really “scalable” or scalable on demand, as implied by the title.

Our intention was not to imply that the cage is scalable “on demand”. Instead, the word “scalable” was chosen to highlight the stepwise structural expansion caused by the increasing number of engineered electrostatic interactions. We recently found that other AaLS variants can adopt an even larger number of discrete sizes/scales than those reported in this manuscript (references 25 and 29), and efforts are currently ongoing to elucidate the general principles underlying these AaLS assemblies with the goal of being able to design differently sized cage structures rationally.

Reviewer #3 (Remarks to the Author):

This is a study of large complexes of the capsid-forming enzyme AaLS employing native mass spectrometry and cryo-electron microscopy to characterize the cage-like structures of these macroassemblies. Three mutants are investigated: the wild type AaLS-wt; an engineered mutant AaLS-neg with four mutations to negatively charged glutamates that project into the lumen of the capsid; and another variant, AaLS-13, obtained by directed evolution and featuring an additional seven mutations which further increase the negative charge in the container. It is found that AaLS-wt forms a 60-mer complex in the overall shape of an icosahedron, that AaLS-neg forms a 180-mer with tetrahedral symmetry, and that AaLS-13 forms a 360-mer in the shape of a dodecahedron. The cage structures are stable up to 80 degrees Celsius. It is explained how an increase of negatively charged residues on one side of the monomer structure leads through Coulomb repulsion to a change in the monomer structure which in turn leads to a decrease in the curvature of assembled oligomers which finally leads to larger oligomer assemblies.

The paper is generally well written, the research is outstanding, and the topic is very interesting to a broad audience of chemists. I recommend publication of this article after a few minor changes.

The reviewer’s strong support is much appreciated.

Detailed comments:

1.) Abstract, line 17: “60-subunit dodecahedron”. For the general audience not intimately familiar with virus and similar structures, it is odd to make a connection between 60 units and the shape of a dodecahedron. Later, in the text of the paper it becomes obvious how this geometry is assembled but in the abstract it is confusing. For the expert this is not an issue, of course, but I would like to bring this point to the attention of the authors and perhaps there is a simple way to alleviate the mismatch between the numbers 12 (shape) and 60 (units).

To address the reviewer’s point, we replaced “a 60-subunit dodecahedron” in the abstract with “a dodecahedron composed of 12 pentamers”. Similarly, we replaced “180 and 360 subunits” with “36 and 72 pentamers” in our description of the engineered variants.

2.) The first paragraph of the “Single-particle reconstruction” section, lines 76-84, can only be understood by experts familiar with the procedure. The authors should make an attempt to rephrase this paragraph to make it more readable for a general audience without expanding too much, though. An example of a very successfully explained complex procedure is the brief explanation of “directed evolution” in one sentence (lines 41-43).

We rewrote the section on single-particle reconstruction (pages 4-5) to make it more accessible to a general reader and also added a recent review describing the procedure (ref. 32).

3.) Lines 123-124: There is a fraction of an incomplete sentence, perhaps a leftover of a draft version. Also, it should say “using atomic force microscopy” or “using an atomic force microscope” instead of “using atomic force microscope”.

Corrected.

4.) Line 140: “... the wedge angle of the pentamer...”. I was not immediately able to understand which angle is referred to. I had to study all figures carefully and read the text in its entirety before I could make sense of this phrase. (It’s an angle perpendicular to the plane of the pentamer pentagon. Initially, I tried to see the pentagon shape itself as a wedge which simply did not make any sense in the following discussion.) I think a reference to figure 6c would be appropriate here: “... the wedge angle of the pentamer (Fig. 6c)...”.

The individual pentamers do have wedge-like shapes. The phrase “wedge angle” in the original text referred to the angle subtended by the short edge of the pentamer relative to the perpendicular, which decreases in the order AaLS-wt > AaLS-neg > AaLS-13 as shown by the dotted lines in Fig. 6b. To avoid confusion, however, we altered the phrase “...reduces the wedge angle of the pentamer...” to “...modifies the wedge-like shape of the pentamer (Fig. 6b)”. We also reworded the following two sentences on page 7-8, which now read “Repositioning of the α 4 helix...further modifies the shape of the pentamer wedge” and “The modified pentamer wedges (Fig. 6b) would cause...” In addition, Supplementary Fig. 6a,b are explicitly called out after the phrase “Repositioning of the α 4 helix (residues 120-131)”.

5.) Line 164: “The position of one protomer is rotated $\sim 20^\circ$ relative to its neighbor, increasing the curvature of the cage while still maintaining contacts in the hinge region (Supplementary Fig. 9).” Increasing the curvature by going from which mutant to which other mutant? A better way would be to replace “increasing” by “changing”: “The position of one protomer is rotated $\sim 20^\circ$ relative to its neighbor, thereby changing the curvature of the cage while still maintaining contacts in the hinge region (Supplementary Fig. 9).”

Modified as suggested.

6.) Line 283: “... one protomer... was adjusted to the EM density using O...”. What is “O”?

O is a software program, which we now note explicitly on page 13.

Reviewer #1 (Remarks to the Author):

The revisions adequately address the concerns raised during review - the paper is now suitable for publication.

Reviewer #2 (Remarks to the Author):

The revised manuscript successfully addresses all concerns raised.